# Oxalate Homeostasis in Non-Stone-Forming Chronic Kidney Disease: A Review of Key Findings and Perspectives

**DOI:** 10.3390/biomedicines11061654

**Published:** 2023-06-07

**Authors:** Natalia Stepanova

**Affiliations:** State Institution «Institute of Nephrology of the National Academy of Medical Sciences of Ukraine», 04050 Kyiv, Ukraine; nmstep88@gmail.com; Tel.: +38-09-6555-1555

**Keywords:** oxalate, chronic kidney disease, hemodialysis, peritoneal dialysis, plasma oxalic acid, hyperoxaluria, cardiovascular disease, gut microbiota, transporters, treatment

## Abstract

Chronic kidney disease (CKD) is a significant global public health concern associated with high morbidity and mortality rates. The maintenance of oxalate homeostasis plays a critical role in preserving kidney health, particularly in the context of CKD. Although the relationship between oxalate and kidney stone formation has been extensively investigated, our understanding of oxalate homeostasis in non-stone-forming CKD remains limited. This review aims to present an updated analysis of the existing literature, focusing on the intricate mechanisms involved in oxalate homeostasis in patients with CKD. Furthermore, it explores the key factors that influence oxalate accumulation and discusses the potential role of oxalate in CKD progression and prognosis. The review also emphasizes the significance of the gut–kidney axis in CKD oxalate homeostasis and provides an overview of current therapeutic strategies, as well as potential future approaches. By consolidating important findings and perspectives, this review offers a comprehensive understanding of the present knowledge in this field and identifies promising avenues for further research.

## 1. Introduction to Health Oxalate Homeostasis

Oxalic acid (C_2_H_2_O_4_) and its anionic form (C_2_O_4_^2−^), collectively known as oxalate, are organic compounds that occur naturally in various plant-based foods, such as spinach, rhubarb and beet greens [1,2,3]. These compounds are also produced by the human body as a byproduct of normal metabolism and are present in small amounts in body fluids, such as urine and plasma [1,2,4].

Under physiological conditions, oxalate homeostasis is maintained through a complex interplay between dietary oxalate intake, oxalate metabolism and absorption in the gastrointestinal tract, and oxalate excretion through the kidneys and intestine (Figure 1) [1,2,4,5].

Under physiological conditions, the total amount of oxalate in the body is composed of endogenous oxalate (60–80%) and exogenous oxalate (20–40%), with about 10–15% of the oxalate originating from the gastrointestinal tract, of which a small fraction (2–10%) is absorbed, while most (90–98%) is used as an energy source by the intestinal microbiota or excreted with feces (5–10%). However, most of the oxalate in the blood is excreted by the kidneys (90–95%). Modified from [1,2,4,5].

### 1.1. Dietary Oxalate Intake

Dietary oxalate intake is critical for maintaining oxalate homeostasis [1,4,6]. It has been demonstrated that around 20–40% of urinary oxalate comes from dietary sources, and this percentage can increase significantly when calcium consumption is reduced [7].

However, only 10–15% of ingested oxalate is absorbed into the systemic circulation through the gastrointestinal tract [8]. Studies have demonstrated that after dietary oxalate intake, there is a temporary increase in plasma oxalic acid (POx) levels, which reach their peak within 2–4 h [9], and more than 75% of the ingested oxalate is excreted within 6 h after the load [8]. This suggests that oxalate intake has a limited impact on POx concentrations and underlines the significance of endogenous production in the liver and absorption from the gastrointestinal tract as the primary sources of the POx burden.

### 1.2. Endogenous Oxalate Metabolism

Endogenous oxalate metabolism is a complex process that involves multiple enzymes and pathways accounting for approximately 60–80% of total oxalate handling in the body [1,2,4]. The liver is the main organ responsible for endogenous oxalate production, with glyoxylate and ascorbic acid being precursor molecules [2,4,5,7]. Around 40% of total endogenous oxalate is synthesized from the breakdown of various amino acids, including glycine, hydroxyproline, and hydroxylysine, through a precursor molecule called glyoxylate, and approximately 30% is derived from the metabolism of ascorbic acid [4,7,10]. The enzyme alanine-glyoxylate aminotransferase (AGT) converts glyoxylate into glycine, which can then be used for protein synthesis or further metabolized to produce energy [10]. A genetic mutation can cause a deficiency in AGT, resulting in the accumulation of glyoxylate and subsequent conversion to oxalate, leading to the development of primary hyperoxaluria [11]. Ascorbic acid is metabolized to oxalic acid through a series of chemical reactions, starting with the conversion of ascorbic acid to L-gluconolactone by the enzyme L-gluconolactone oxidase. L-gluconolactone is then converted to L-guano-1,4-lactone by the enzyme L-guano-1,4-lactone oxidase, which is further metabolized to oxalate by a series of enzymatic reactions involving glycolate oxidase, glyoxylate reductase, and lactate dehydrogenase [7,12].

### 1.3. Intestinal Oxalate Absorption and Bacterial Degradation

Oxalate absorption in the gastrointestinal tract is a complex process that involves both passive (paracellular) and active (transcellular) mechanisms [2,4]. The majority of oxalate absorption occurs in the small intestine, specifically in the proximal jejunum, with a smaller amount absorbed in the colon [4,13]. Paracellular transport of oxalate refers to the movement of the oxalate anion between epithelial cells, driven by the concentration and electrical gradients across the cell layers and the properties of the tight junctions, without requiring any energy input [4,13]. This process is influenced by the concentration gradient of oxalate, the pH of the gastrointestinal contents, and the surface area of the intestine available for absorption [1,4]. Transcellular absorption, on the other hand, requires the input of energy to move oxalate against its concentration gradient. This process involves specific transporters that move oxalate from the intestinal lumen into enterocytes [13]. The SLC26A family of anion exchangers, specifically SLC26A6 in the small intestine and SLC26A3 in the colon, play significant roles in the transcellular transport of oxalate [2,14,15]. SLC26A6 is expressed in the apical membrane of the small intestine and facilitates the exchange of bicarbonate ions for oxalate ions, promoting the secretion of oxalate from the bloodstream into the intestinal lumen. This process helps eliminate excess oxalate from the body [2,14]. It has been demonstrated that mice lacking SLC26A6 exhibit significant impairments in the intestinal release of oxalate, which may increase the absorption of net oxalate, ultimately resulting in hyperoxalemia [4,13]. SLC26A3, also known as the downregulated in adenoma (DRA) protein or chloride/bicarbonate exchanger, is primarily responsible for the absorption of oxalate in the intestine [16]. It is predominantly expressed in the apical membrane of the colon and plays a crucial role in the reabsorption of oxalate from the intestinal lumen into the bloodstream [2,4,13,16,17].

Recent findings suggest that not only *Oxalobacter formigenes* (*O. formigenes*), but also various other oxalate-degrading bacteria (ODB), including *Enterococcus* spp., *Lactobacillus* spp., *Bifidobacterium* spp., and *Bacillus* spp., can break down oxalate, reducing its absorption and urinary excretion by 40% [18,19]. It is important to note that oxalate-degrading activity (ODA) is not limited to the aforementioned species but rather is shared among a large number of taxa influencing each other in a complex metabolic network [19,20,21]. The primary mechanism by which ODB metabolizes oxalate is through the expression of oxalate decarboxylase (OxdC), an enzyme that converts oxalate into formate and carbon dioxide [18,19,20]. Some ODB also produce an oxalyl-CoA decarboxylase (Oxc), which can further degrade oxalate into glyoxylate and formate [18,20]. The formate produced during this process can be utilized by bacteria as an energy source. Additionally, some ODB express other enzymes, such as formyl-CoA transferase and formate dehydrogenase, which participate in the metabolism of formate [18]. By degrading oxalate in the gut, ODB can prevent the absorption of oxalate into the bloodstream and maintain oxalate balance in the body [18,20]. It should be noted that oxalate absorption in the gut has significant intraindividual variability and can be influenced by several factors [13,18]. These include the abundance of ODB in the gut and calcium and fiber intake. High dietary calcium intake can decrease oxalate absorption by forming insoluble calcium oxalate in the gut, while low calcium intake can increase oxalate absorption [22,23]. Similarly, high fiber intake can reduce oxalate absorption by binding with oxalate and increasing fecal excretion, whereas low fiber intake can increase oxalate absorption [18,24]. Other factors such as gut motility, pH, antibiotics, and other medications used can also affect oxalate absorption in the gut [18,25,26,27].

### 1.4. Oxalate Excretion

As humans lack the ability to degrade oxalate, oxalate excretion through the kidneys and intestine is critical to maintaining oxalate homeostasis. The kidneys play a vital role in eliminating the majority (90% to 95%) of circulating oxalate, while approximately 5% to 10% of blood oxalate is excreted with feces [2,4,5]. In the kidneys, oxalate is filtered through the glomeruli and is then reabsorbed or secreted by tubular epithelial cells [28]. SLC26A6 and other transporters, such as SLC26A1, SLC22A8, and SLC22A12, are involved in the renal excretion of oxalate. These transporters are responsible for the uptake of oxalate from the blood into renal tubular cells, where it is then excreted into the urine [2,4,13]. Overexpression of SLC26A6 in the kidneys has been shown to increase oxalate excretion and urinary oxalate (UOx) concentrations in humans [29]. The regulation of plasma oxalate (POx) levels depends not only on glomerular filtration but also on the critical contribution of tubular secretion. There is a strong correlation between a high POx concentration and oxalate secretion, highlighting the importance of both processes in maintaining oxalate homeostasis [30].

The intestine also serves an essential role in the elimination of oxalate from the body. In the intestine, oxalate binds with calcium to form a compound that cannot be absorbed passively through tight junctions in the intestinal epithelium, resulting in its excretion through feces [2,4,5,13]. However, in physiological conditions, calcium levels are insufficient, and oxalate, water, and salt can easily pass through tight junctions. Unabsorbed oxalate is excreted in feces, whereas absorbed oxalate may be transported back to the liver via enterohepatic circulation [5,13,31]. In addition to calcium levels, dietary oxalate intake, the composition of the gut microbiota, and the efficiency of intestinal oxalate absorption may affect the fecal excretion of oxalate [13].

## 2. Oxalate Implications in Health and Disease

Oxalate is often referred to as an “anti-nutrient” because of its ability to bind with minerals and form various solubility complexes. Specifically, oxalate is a negatively charged ion that forms crystals by binding with positively charged ions, such as calcium, magnesium, iron, zinc, sodium, and potassium [32,33]. The solubility of these salts varies widely, with sodium and potassium oxalate being soluble and calcium, magnesium, zinc, iron, and other cations forming as less soluble to practically insoluble oxalate compounds [33]. Specifically, the solubility product constant (Ksp) values at 25 °C for these compounds are as follows: calcium oxalate (CaC_2_O_4_) with a Ksp of 2.7 × 10^−9^, magnesium oxalate (MgC_2_O_4_) with a Ksp of 8.5 × 10^−5^, zinc oxalate (ZnC_2_O_4_) with a Ksp of 2.7 × 10^−8^, and iron(II) oxalate (FeC_2_O_4_) with a Ksp of 2 × 10^−7^ [34]. This binding capacity of oxalate with minerals can cause mineral deficiencies and is associated with the formation of calcium oxalate kidney stones, thus earning the term “anti-nutrient” [33,35].

Oxalate is synthesized by a variety of cells, including liver cells, kidney cells, epithelial cells, and apocrine cells [3,36]. However, despite being a byproduct of cellular metabolism, the physiological functions of oxalate in the human body remain largely unknown. Although it was traditionally believed that circulating oxalate has no function in the human body, early studies have described three potential physiological roles. First, oxalate plays a role in the proximal tubule of human kidneys, where it is transported by the membrane transporter SLC26A6. This transport process has been found to stimulate the absorption of chloride, water, and sodium [14,37]. The second role is related to the production of H_2_O_2_ by oxalate oxidase, which can enhance the burst of phagocytes [38]. Finally, uracil and orotic acid, which are essential components of RNA involved in protein synthesis and pyrimidine nucleotide synthesis, respectively, require oxalate for their formation in human metabolism [3].

In trace amounts, oxalate is generally considered physiologically inert and is excreted from the body without clinical significance [4,5]. However, a decrease in glomerular filtration rate (GFR), as well as an increase in hepatic oxalate production or gastrointestinal oxalate absorption, can raise POx concentrations and result in increased UOx excretion, thereby elevating the risk of various pathological conditions [4,5,39]. Calcium-oxalate (CaOx) urolithiasis, primary hyperoxaluria, and oxalate nephropathy caused by either high oxalate intake or enteric hyperoxaluria are among the most thoroughly researched conditions [33,39,40]. Besides, elevated plasma and/or urinary oxalate have been linked to a number of other pathologic conditions, such as diabetes mellitus and obesity [41], autism [42], atherosclerosis [43], cardiovascular events [44], and neurological disorders [39]. Patients with primary hyperoxaluria are especially prone to oxalosis, a condition where CaOx deposits have been found in various extrarenal tissues, primarily in the heart, smooth muscle cells of vessels, bones, skin, and other organs [45]. However, oxalate deposition has also been reported in non-primary hyperoxaluria patients affecting extrarenal tissues such as the breast [46], lungs [47], thyroid gland [48], prostate [49], synovial fluid [50], and vascular tissues [51]. Furthermore, studies have suggested that oxalate may promote the proliferation of cancer and metastatic cells, contributing to the development of breast and prostate cancer [36,52,53]. The diversity of pathological conditions associated with oxalate, notwithstanding the vital role played by the kidneys and intestine in maintaining oxalate homeostasis, underpins the significant oxalate implications in the development and progression of chronic kidney disease (CKD).

## 3. The Interplay between Oxalate and CKD: A Vicious Cycle of Shared Risk Factors

CKD is a significant public health problem worldwide, as affected individuals are at increased risk for end-stage kidney disease (ESKD), cardiovascular disease, and a wide range of CKD-related mental and physical illnesses, leading to premature mortality [54,55]. Oxalate has been shown to be associated with CKD progression [56,57], CKD- and ESKD-associated cardiovascular diseases [58,59,60,61], polycystic kidney disease progression [62], and/or poor renal allograft survival [62,63]. In CKD of any cause, impaired kidney function and associated gut dysbiosis can cause a buildup of plasma and glomerular ultrafiltrate oxalate, increasing oxalate exposure of the remaining tubule cells and decreasing the renal clearance of oxalate, leading to its accumulation in the body [4,5,28]. Oxalate retention can initiate a vicious cycle of progressive kidney damage and a further decline in GFR. In turn, the formation of oxalate crystals in the kidneys can accelerate inflammation and scarring, leading to a decline in kidney function and contributing to the progression of CKD [56,64]. This interplay forms a vicious cycle between oxalate and CKD with shared risk factors, where each condition exacerbates the other by promoting the accumulation of oxalate in the body, ultimately leading to an increased risk of CKD progression (Figure 2).

Specifically, it has been observed that an imbalanced intake of oxalate and calcium in the diet is associated with a significant increase in the risk of developing hypertension [65], which is a strong risk factor for both impaired oxalate homeostasis and CKD through its damaging effect on the renal microvasculature. In obesity, there is an increased risk of developing both insulin resistance and hypertension, which can contribute to impaired oxalate homeostasis and CKD [41,66]. Similarly, metabolic syndrome, which includes insulin resistance, dyslipidemia, hypertension, and obesity, is independently associated with both CaOx nephrolithiasis and CKD [67,68]. Furthermore, a mutation in the SLC26A6 oxalate transporter has been linked to enteric hyperoxaluria and nephrolithiasis, indicating the possibility of an inherited form of this condition [69]. Although there is limited information on the relationship between SLC26A6 mutation and CKD, this anion transporter plays an important role in kidney salt absorption, acid-base balance, vascular volume homeostasis, and blood pressure regulation, all of which contribute to the advancement of CKD [70]. Acidic urine pH (<5.0) is also associated with significantly increased UOx and protein excretion, resulting in CKD progression [71,72]. While it is generally believed that urine pH has little impact on the solubility of CaOx, some studies have shown that CaOx monohydrate crystals are likelier to form at pH 4.0 and less likely to form at pH 8.0, indicating a potential influence of urinary pH [73]. Moreover, changes in urinary pH can affect the excretion of specific substances such as citrate, which has been found to inhibit the formation of CaOx. Consequently, alterations in urinary pH can disrupt the balance of citrate, potentially leading to hyperoxaluria and contributing to the progression of CKD. Notably, CKD often presents with hypocitraturia and/or metabolic acidosis, which are characterized by low urine pH [74]. As a compensatory mechanism for the loss of nephrons and impaired overall acid excretion, there is an increase in acid excretion per nephron, which further promotes damage to the tubulointerstitial region and contributes to the progression of kidney disease [74].

Finally, dysbiosis of the gut microbiota (described below), as well as the use of nephrotoxic medications or medications that affect the gut microbiome and metabolome, have been shown to contribute to both increased oxalate burden and CKD [21,26,75]. Proton pump inhibitors, for example, are known to cause intestinal permeability, leading to both urolithiasis and CKD progression [76,77]. Antibiotics, high-risk nephrotoxicity drugs, can alter the gut microbiota, reducing the abundance of either ODB or their ability to degrade oxalate, increasing the risk of oxalate burden [21,26]. Similarly, vitamin D deficiency, a nearly common feature in CKD, was also shown to impact gut microbiota diversity, causing intestinal barrier dysfunction [78]. Additionally, a deficiency in vitamin D may lead to elevated levels of parathyroid hormone, which can stimulate bone resorption and cause hypercalcemia and hypercalciuria, potentially increasing the risk of hyperoxaluria [79].

## 4. Oxalate’s Role in the Pathogenesis of CKD: From Silent Culprit to Active Player

While oxalate has long been known to be a risk factor for kidney stones, its involvement in the pathogenesis of CKD has only recently come into focus. For many years, oxalate was considered a “silent culprit” in CKD, with its effects on kidney function thought to be limited to the formation of kidney stones. The primary mechanism for CKD from kidney stones is usually attributed to obstructive uropathy or recurrent kidney stone disease experience [80,81]. In fact, oxalate may play a much more active role in the development and progression of CKD, contributing to kidney damage beyond the formation of kidney stones.

The mechanisms involved in kidney damage caused by oxalate are multifactorial and complex, and their complete understanding is still lacking. The primary way by which oxalate can contribute to CKD is through the impairment of mitochondrial function [82]. Mitochondria are energy-producing organelles within cells, and disruption of mitochondrial function has been implicated in a number of disease processes, including CKD [83,84]. Oxalate can trigger mitochondrial dysfunction in renal epithelial cells through various mechanisms, such as impairing the mitochondrial respiratory chain, increased reactive oxygen species (ROS) generation and inflammation, disrupting the mitochondrial membrane potential, and affecting mitochondrial biogenesis [82,85,86]. In endothelial cells, POx at uremic concentrations alters intracellular calcium levels, increases the production of ROS, and promotes cell apoptosis, leading to oxidative stress and inflammation [87,88]. Oxalate-induced oxidative stress and inflammation lead to the activation of various signaling pathways, such as nuclear factor kappa B (NF-κB), mitogen-activated protein kinases (MAPKs), and NLRP3 inflammasome, and promote the production of proinflammatory cytokines and chemokines, such as interleukins (IL) -1β, -6, tumor necrosis factor α (TNF-α), monocyte chemoattractant protein-1 (MCP-1), and transforming growth factor β1 (TGF-β1) in renal epithelial cells [1,60,87,89]. In turn, the activation of the TGF-β signaling pathway induces extracellular matrix (ECM) synthesis and deposition, which can lead to fibrosis and scarring in the kidneys [90,91]. Oxalate can increase MCP-1 and TGF-β expression and activate downstream signaling pathways, leading to the upregulation of ECM proteins, such as collagen and fibronectin [90,91]. In addition, oxalate can induce apoptosis of renal epithelial cells through the activation of various apoptotic signaling pathways, such as caspase-3 and -9, and the Bcl-2 family of proteins [92].

At the systemic level, oxalate has been shown to stimulate human monocytes and promote the production of proinflammatory cytokines such as TNFα, IL-1β, IL-6, and IL-8 in vitro [93], leading to systemic low-grade inflammation, which is a significant contributor to the progression of CKD [94,95,96]. Additionally, oxalate can compromise macrophage metabolism, disrupt redox homeostasis, and alter cytokine signaling, resulting in a reduced antibacterial response and increased risk of infection [89]. Altered oxalate metabolism has also been shown to promote atherosclerosis by dysregulation of redox status, enhance inflammatory response, and affect cholesterol metabolism both in vitro and in vivo using genetically modified mice lacking the alanine-glyoxylate aminotransferase enzyme and the apolipoprotein E gene (apoE^−/−^) [43]. In the CKD model induced by a high-oxalate diet in C57BL/6 mice, the majority of CKD-related complications have been observed, including mineral bone disease, dyselectrolytemia, metabolic acidosis, arterial hypertension, and even cardiac fibrosis [60]. In the uremic atherosclerosis model using apoE^−/−^ mice, a significant increase in aortic oxalate levels and serum levels of oxidative stress and inflammatory markers have been observed, suggesting a significant role for hyperoxalemia in promoting oxidative stress and systemic inflammation [87].

Overall, it is becoming increasingly clear that oxalate may play a much more active role in the pathogenesis of CKD beyond its traditional association with kidney stone formation. Oxalate can directly impact renal epithelial cells, activate various signaling pathways involved in inflammation, fibrosis, and apoptosis, and contribute to systemic low-grade inflammation. While the current understanding of the mechanisms by which oxalate contributes to CKD has improved, further research is necessary to fully comprehend its impact on kidney health.

## 5. Gut-Kidney Axis in CKD Oxalate Homeostasis

The gut–kidney axis is essential in maintaining oxalate homeostasis in CKD [5,31,64,97]. Despite extensive research, the precise mechanisms by which the kidneys and gut communicate to regulate oxalate homeostasis in CKD remain incompletely understood [31]. Nonetheless, it is evident that various conditions associated with secondary hyperoxaluria, such as Crohn’s disease, chronic pancreatitis, Roux-en-Y gastric bypass, and antibiotic use, contribute to the progression of kidney disease [28,45]. These conditions are linked to alterations in the gut microbiota and enteric oxalate handling, disrupting oxalate homeostasis and leading to hyperoxaluria/hyperoxalemia [5,18,31,98]. For example, patients with inflammatory bowel disease have been shown to have elevated enteric oxalate levels and reduced oxalate-degrading gene expression caused by the loss of *O. formigenes*, resulting in hyperoxaluria [19].

CKD, on the other hand, is a widely discussed cause of gut microbiota dysbiosis characterized by an increase in proteolytic bacterial populations, which produce uremic toxins such as ammonia, phenols, and indoles, and a decrease in saccharolytic populations that form short-chain fatty acids (SCFAs) [97,99]. CKD-associated gut microbiota dysbiosis can impact oxalate homeostasis directly (via a decrease in bacterial ODA) or indirectly (via compromising intestinal barrier function). In our previous report, we showed that ODA in fecal microbiota has a direct association with the percentage of renal interstitial fibrosis in rats following glycerol-induced acute kidney injury over 70 days [100]. In addition, total fecal ODA has been found to be significantly lower in patients with ESKD compared to healthy controls [101,102]. Interestingly, it appears that ODA, rather than the quantity of ODB in the gut microbiota, may have an impact on oxalate homeostasis [21,102]. This hypothesis was supported by findings on the inverse association between ODA in fecal microbiota and serum indoxyl sulfate, UOx, and POx concentrations [101,102]. However, further investigation is required to confirm these findings.

Intestinal dysbiosis and compromised intestinal barrier function can facilitate the translocation of bacterial products and uremic metabolites from the gut lumen to the bloodstream, contributing to oxidative stress, chronic inflammation, and CKD progression [99,103,104]. Uremic metabolites such as indoxyl sulfate, p-cresyl sulfate, and their precursors have been shown to directly promote CaOx crystal production in vitro [105], while SCFAs reduced UOx excretion possibly by the regulation of the expression of SLC26A3/6 transporters [106]. Notably, the SLC26A3 transporter, a crucial chloride-bicarbonate exchanger responsible for the reabsorption of oxalate from the intestinal lumen into the bloodstream, plays a significant role in maintaining the integrity of the intestinal barrier [107]. The deletion of the DRA protein, an isoform of the SLC26A3 transporter, in knockout mice has been found to lead to a notable decrease in UOx and POx concentrations [108].

The presence of chronic inflammation caused by dysbiosis further exacerbates the decline in GFR, leading to reduced renal clearance of oxalate and the development of hyperoxalemia. In response to hyperoxalemia, enteric oxalate excretion becomes an important compensatory mechanism for maintaining oxalate homeostasis [2,13,64]. It has been hypothesized that in CKD, the intestine may adapt and increase the secretion of oxalate into the intestinal lumen by upregulating the expression of oxalate transporters [15,64]. A recent study on animal models of CKD found that the SLC26A6 oxalate transporter, which plays a key role in restricting net absorption by back-secreting oxalate into the lumen, is upregulated not only in the small intestine but also in the colon, resulting in increased oxalate removal through fecal excretion [15]. Controversially, reduced fecal oxalate excretion and increased POx concentration were observed in SLC26A6 gene-deficient mice, suggesting that enteric oxalate secretion via SLC26A6 plays a vital role in defending against hyperoxalemia in CKD [15]. It is essential to note that this is the only study on intestinal SLC26A6 expression in CKD models, and much work remains to be done to improve our understanding of the impact of the gut-kidney axis on oxalate homeostasis in CKD. Although a definitive understanding of this complex relationship remains incomplete, Figure 3 provides a schematic summary of current knowledge on the gut–kidney axis in CKD oxalate homeostasis.

## 6. Oxalate as a Clinical Marker for CKD Progression and Prognosis

Increased oxalate concentrations in both plasma and urine are commonly observed but often overlooked complications in patients with CKD. Hyperoxaluria (≥45 mg/24 h) is estimated to affect 5–24% of patients with gastrointestinal disease associated with malabsorption [109]. Although it is unclear how common hyperoxaluria is in CKD, it could have a significant impact, affecting approximately 250,000 individuals with gastrointestinal disease in the United States in 2019, which may potentially lead to kidney failure [110]. The prevalence of hyperoxalemia in CKD patients also remains unclear and has only recently been the subject of a few studies examining POx levels in both the general CKD population [57,111,112] and the dialysis cohort [113]. At reference values of 1–3 µmol/L, POx concentrations in patients with CKD range from 1 to ≥110 μmol/L, depending on estimated GFR (eGFR) [111,112,113]. These studies have shown a progressive increase in POx concentrations with decreasing eGFR, with the highest levels observed in patients undergoing hemodialysis (HD) [111,112,113]. However, despite the clear trend linking oxalate concentrations to eGFR in CKD patients, significant variations in POx and UOx levels have been reported among patients with the same eGFR level [57,112,113]. This issue is particularly exacerbated in anuric patients undergoing HD. Although the patient population is nearly homogeneous and underwent a standardized dialysis regimen, intraindividual POx concentrations vary significantly [102,113,114]. These variations can be attributed to other factors, such as differences in the amount of ingested oxalate, hepatic metabolism and intestinal absorption. For example, the daily UOx excretion increases by 2.7 mg per 100 mg of dietary oxalate ingested [6]. Patients with diabetes have been found to have increased plasma glyoxylate levels, which are intermediate molecules in the metabolic pathway of sugar and oxalate, resulting in increased POx and UOx concentrations [41]. Similarly, gastrointestinal disorders or medication intake may also affect bacterial ODA and oxalate absorption in the gut, determining POx and UOx concentrations in patients with ESKD and anuria [18,26,102].

Although there is ample in vitro and experimental evidence that oxalate plays a role in the development and progression of CKD, few clinical studies have examined the association between plasma or urinary oxalate levels and CKD progression. In a prospective cohort study, Wailkar et al. investigated the association between 24-h UOx excretion and the risk of CKD progression and kidney failure in 3123 participants with grades 2 to 4 CKD [115]. The study found that UOx excretion higher than 27.8 mg/24 h was independently associated with a 32% increased risk of CKD progression and a 37% increased risk of kidney failure [115]. Likewise, in a prospective study involving 167 stable kidney transplant recipients, a POx concentration greater than 13 µmol/L, measured over 10 weeks following transplantation, was found to be significantly associated with impaired long-term patient and graft survival during a 15-year follow-up period [63].

In addition to CKD progression, oxalate has also been linked to cardiovascular complications associated with CKD. Because oxidative stress and chronic inflammation are major risk factors for accelerated atherosclerosis [116,117], they seem to be the pathophysiological link between hyperoxalemia and cardiovascular disease (CVD) [87]. CaOx crystal deposits were observed in coronary arteriosclerotic lesions in patients treated with HD and peritoneal dialysis (PD) [118]. Elevated POx levels were also significantly associated with an increasing trend in atherogenic lipoprotein fractions and proinflammatory markers in patients undergoing dialysis kidney replacement therapy (DKRT) [58]. Moreover, in a prospective observation of 50 patients with ESKD, POx concentration ≥62.9 μmol/L was significantly associated with CVD events during the 2-year follow-up period, independent of other CVD risk factors [58]. Pfau et al. conducted a study in a well-established cohort of patients with type 2 diabetes mellitus undergoing HD, demonstrating a nearly linear increase in the risk of sudden cardiac death per doubling of POx concentration, where a POx concentration ≥59.7 µM was associated with a 40% increase in the risk of cardiovascular events and a 62% increase in the risk of sudden cardiac death, compared to those with a POx concentration ≤29.6 µM [59].

Despite compelling evidence suggesting the involvement of oxalate in CKD development and progression and its potential as a clinical marker, several limitations currently preclude oxalate from being used as a reliable clinical marker for CKD progression. One major limitation is the lack of standardized methods for measuring oxalate concentrations in clinical samples, leading to variability and inconsistency in reported results [57]. Additionally, oxalate levels can be influenced by various factors, such as diet, gut microbiota, and medication use, which can confound the interpretation of results. Furthermore, most of the existing studies on oxalate and CKD have focused on urolithiasis or primary hyperoxaluria populations, which may not accurately reflect the oxalate level and its clinical associations in the broader CKD population. Therefore, while oxalate may have potential as a clinical marker for CKD progression, further research is needed to establish standardized methods for measuring oxalate levels and to determine its role in a broader population of non-stone-forming patients with CKD.

## 7. Targeting Oxalate Homeostasis to Reduce CKD Progression and the Risk of Cardiovascular Events

Despite the increasing awareness of oxalate’s involvement in CKD pathogenesis and its correlation with cardiovascular outcomes, there are currently no approved treatments that specifically target mitigating oxalate burden in CKD patients. This creates a challenge for patients and healthcare providers in managing oxalate-related issues in the context of CKD, especially for those undergoing DKRT. Managing oxalate balance to prevent CKD progression necessitates a multifaceted approach, but only limited treatment options are currently available.

### 7.1. Dialysis Treatment for Management Oxalate Burden in Patients with Kidney Failure

Oxalate, with a weight of 90 Da, meets the criteria for small, water-soluble uremic toxins [119,120]. Like other small, water-soluble compounds, such as urea or uric acid, oxalate can be effectively removed through HD, depending on the type of dialyzer and its surface area [121]. During one dialysis session, POx concentrations can decrease by approximately 90%, approaching levels similar to those found in healthy individuals [114]. However, POx concentrations increase rapidly after dialysis and typically return to predialysis levels within two hours [114]. In PD, oxalate removal is achieved through the combination of dialysate and urine clearance, which is comparable to the 24 h excretion of oxalate in healthy subjects [122,123]. However, the rate of oxalate removal is influenced by the balance between the peritoneal and kidney routes, with peritoneal clearance being more significant than residual kidney function in controlling POx levels [122,123]. This means that even a single episode of peritonitis or other complications that impact the function of the peritoneal membrane can increase POx levels in patients treated with PD [123]. Therefore, although DKRT is the only effective option for significantly reducing POx concentrations, it alone is insufficient to maintain oxalate homeostasis in patients with ESKD. To optimize oxalate removal in these patients, it is necessary to provide adequate dialysis with a targeted dialysis dose and careful management of dialysis-related complications.

### 7.2. Modifying the Shared Risk Factors to Prevent CKD Progression

One of the possible strategies for preventing CKD progression is to modify the risk factors associated with altered oxalate homeostasis. These include lifestyle modifications such as dietary modifications, increased fluid intake, and increased physical activity, as well as the management of underlying conditions such as metabolic syndrome, diabetes, hypertension, and obesity.

Dietary modifications can play an important role in controlling oxalate levels in CKD but are difficult to achieve in practice [6,7,98]. Moreover, although limiting the consumption of oxalate-rich foods and maintaining a balanced calcium diet are crucial for controlling kidney stone formation, the effect of a low-oxalate diet on CKD progression is still uncertain because of the paucity of evidence. The only study on this topic has recently been conducted by Mirmiran et al. [65]. In their large-scale prospective study with a follow-up of more than eight years, the authors demonstrated that higher dietary oxalate intake may increase the risk of developing hypertension and CKD, whereas lower calcium intake may exacerbate the adverse effects of excessive oxalate intake [65]. In addition to a balanced oxalate-calcium diet, a high fluid intake is strongly recommended to prevent CaOx stone disease [124,125]. However, hypervolemia is common in patients with CKD, and fluid restriction may be necessary. Increased physical activity may also help to achieve a favorable oxalate balance and reduce the risk of hyperoxalemia by addressing CKD risk factors, such as facilitating weight loss, lowering blood pressure, and improving insulin sensitivity [126,127]. Managing these and other modifiable shared risk factors for both oxalate burden and CKD can mitigate the risk of hyperoxalemia and CKD progression. However, again, there is a general lack of data on this topic.

### 7.3. Medication Adjustments

Although there are no medications specifically approved for targeting the oxalate burden in CKD, it is worth considering some of the medications that are recommended or discussed for treating CaOx urolithiasis and are also commonly used in CKD practice.

Calcium-based phosphate binders. Calcium-based phosphate binders, including calcium acetate and calcium carbonate, are commonly used to prevent mineral-bone disorders associated with CKD and are also considered to prevent CaOx stone formation [23,128]. These supplements work by binding with phosphate in the intestine, thus reducing its bioavailability and absorption, resulting in lower urinary oxalate excretion and a decreased risk of hyperoxaluria and CaOx stone formation [23]. To achieve a neutral calcium balance and avoid the adverse effects of either negative or positive calcium balance, a calcium intake of around 1000 mg/day is recommended [128,129]. However, calcium supplementation, especially if administered between meals, can raise urinary calcium excretion without any positive impact on oxalate, thereby elevating the risk of stone formation [23]. Moreover, caution should be exercised when using calcium supplements in patients with advanced CKD because they may cause hypercalcemia and vascular calcification [128,130]. Therefore, the decision to use calcium-based phosphate binders in patients with advanced CKD should be made on a case-by-case basis, balancing the individual needs and risks of each patient against the risk of worsening vascular calcification.Noncalcium phosphate binders. Noncalcium phosphate binders, such as lanthanum carbonate, have also been shown to reduce UOx excretion in CKD patients by decreasing gut absorption of dietary oxalate [131,132]. This is thought to be due to the ability of lanthanum to form insoluble complexes with oxalate, thus reducing its bioavailability for absorption [131,133]. However, more extensive studies are needed to establish the efficacy and safety of noncalcium phosphate binders in preventing hyperoxalemia/hyperoxaluria and CKD progression.Calcium channel blockers. Verapamil has been shown to increase urinary oxalate excretion and reduce the risk of CaOx stone formation in animal models [134]. However, their efficacy in patients with hyperoxalemia/hyperoxaluria has not been established, and the risk-benefit ratio for the individual patient should determine the treatment option.Thiazide and thiazide-like diuretics. Thiazide diuretics have been found to be effective in treating hypertension in patients with ESKD [135] and in reducing the formation of CaOx kidney stones. Thiazide-type diuretics (hydrochlorothiazide, chlorthalidone, and indapamide) act on the distal tubule of the kidney, increasing calcium reabsorption and decreasing the excretion of calcium in the urine, resulting in a decrease in UOx excretion [136,137]. However, the use of thiazide and thiazide-like diuretics in CKD patients is often restricted due to concerns regarding their safety and effectiveness, such as the risk of electrolyte imbalances, volume depletion, and a decline in eGFR [138].Magnesium. Magnesium has been recognized as a potent inhibitor of calcium oxalate (CaOx) crystals due to its ability to bind with oxalate, forming a soluble complex. This inhibitory effect is particularly significant when magnesium is combined with citrate and remains effective even in acidic environments [139]. Magnesium also inhibits the absorption of dietary oxalate from the gut lumen, as well as citrate-rich food [140]. Furthermore, magnesium plays a multifaceted role in the management of CKD. It has been shown to suppress the secretion of parathyroid hormones, activate the calcium-sensing receptor, promote osteoblast activity, and reduce intestinal phosphate absorption. These mechanisms contribute to the regulation of mineral metabolism and prevent the development of secondary hyperparathyroidism and vascular calcification in CKD patients. Additionally, magnesium has been associated with a decrease in the incidence of vascular calcification and improvements in cardiac function [141,142,143]. Incorporating magnesium supplementation as a medical adjustment in patients with CKD may not only help address oxalate burden but also maintain mineral balance, reduce the risk of complications, and improve overall cardiac health. However, further studies are needed to determine optimal dosing strategies and assess the long-term effects of magnesium supplementation in CKD populations.Vitamins B6 and D. Extensive research has been conducted on the association between deficiencies in vitamins B6 and D and the development of CaOx urolithiasis [23,144,145,146]. However, the effectiveness of vitamin supplementation in preventing hyperoxalemia/hyperoxaluria is still a controversial issue [122,145,146]. Additionally, there is a lack of knowledge regarding the potential of these vitamins to prevent the oxalate burden in CKD patients. More research is required to ascertain the potential benefits of these vitamins in managing disrupted oxalate homeostasis in CKD.

### 7.4. Enhancement of Intestinal Oxalate Handling with Promising New Pharmacological Targets

Excitingly, in recent times, novel pharmacological targets have emerged that may provide new possibilities for the treatment of oxalate-related disorders. These pharmacological targets focus on improving extrarenal oxalate clearance, opening a promising avenue for the development of innovative therapies.

ODB. ODB have been extensively studied as a potential treatment option for patients with hyperoxaluria and urolithiasis [147]. By degrading oxalate in the gut, ODB may prevent the absorption of oxalate into the bloodstream and reduce the burden of oxalate in CKD patients. Studies have shown that probiotics and synbiotics can be considered good sources of naturally occurring oxalate-degrading agents in the human colon. Pro- and/or synbiotics supplements containing *O. formigenes, Bifidobacterium lactis*, Lactobacillus strains, and others have been found to decrease hyperoxalemia/hyperoxaluria [21,147,148,149,150], but the results of in vitro and experimental studies do not always reflect the ability of bacteria to degrade oxalate in humans [18,147,151]. Specific clinical studies are scarce, and further research is needed to determine the optimal dosages and benefits of ODB supplementation in the management of the oxalate burden in CKD.Oxalate-degrading enzymes. Oxalate-degrading enzymes represent a new class of enzymes that can effectively degrade oxalate into nontoxic compounds [150,151]. As described above, enzymes, such as OxdC and Oxc, have been found naturally in the gut microbiota [18]. Two promising oxalate-degrading enzymes, reloxaliase and Oxazyme^®^, are currently under investigation as potential treatments for oxalate-related diseases. Reloxaliase, also known as ALLN-177, is a recombinant OxdC enzyme derived from Bacillus subtilis and expressed in Escherichia coli and developed for the treatment of enteric hyperoxaluria [152]. Clinical trials have demonstrated the safety and efficacy of reloxaliase, showing a significant reduction in UOx levels in patients with enteric hyperoxaluria [152,153]. Notably, reloxaliase has also shown promising results in lowering urine and plasma oxalate in patients with CKD, including those with moderate to severe kidney dysfunction. Specifically, it has led to a ~30% decrease in UOx in two patients with grade 3b CKD and a similar reduction in POx in seven patients with grade 5 CKD [154]. Oxazyme^®^ is a synthetic enzyme engineered to effectively degrade oxalate in the gastrointestinal tract [155]. Although clinical trials for Oxazyme^®^ are still in the early stages, preclinical studies have demonstrated its ability to degrade oxalate in laboratory settings [155]. Further research and clinical trials are needed to establish their efficacy, safety, and optimal dosing regimens for the CKD patient population.Small-molecule inhibitor. A small-molecule inhibitor of the intestinal anion exchanger SLC26A3 has been identified for the treatment of hyperoxaluria [151]. The small-molecule SLC26A3 inhibitor (DRAinh-A270) selectively inhibits SLC26A3-mediated chloride/bicarbonate exchange and oxalate/chloride exchange [13,17]. In colonic closed loops in mice, luminal DRAinh-A270 inhibited oxalate absorption by 70% [17]. By selectively inhibiting SLC26A3-mediated oxalate absorption, this inhibitor has the potential to alleviate the burden of oxalate-related complications and improve the management of hyperoxaluria. Continued research and clinical investigations are necessary to fully explore the therapeutic potential of this small-molecule inhibitor and advance its translation into clinical practice.

## 8. Conclusions and Future Directions

Oxalate homeostasis in CKD is a complex process influenced by various factors, including kidney function, gut microbiota, enteric oxalate handling, and intestinal barrier function. Impaired kidney function and CKD-associated gut dysbiosis contribute to the accumulation of oxalate in the body, establishing a detrimental cycle that promotes further oxalate accumulation and increases the risk of CKD progression. The relationship between oxalate homeostasis and CKD is further complicated by shared risk factors, such as diabetes, hypertension, obesity, and metabolic syndrome. CKD itself can disrupt oxalate homeostasis, leading to elevated oxalate levels. Conversely, elevated oxalate levels can contribute to the development and progression of CKD. This interrelationship emphasizes the importance of understanding the role of oxalate in CKD beyond its well-established associations with urolithiasis and primary hyperoxaluria.

The precise mechanisms through which oxalate contributes to the progression of CKD are not yet fully understood. However, one significant mechanism that is thought to play a role is the generation of ROS induced by oxalate. ROS-induced damage to cells and tissues triggers inflammatory responses, which in turn, contribute to the progression of CKD and increase the risk of cardiovascular complications.

Furthermore, the complex relationship between gut microbiota, intestinal barrier function, and intestinal oxalate handling plays a crucial role in maintaining oxalate homeostasis. CKD not only directly affects the abundance and function of ODB but also influences the expression and activity of oxalate transporters involved in intestinal absorption and secretion processes. As a compensatory response, CKD may lead to increased fecal excretion of oxalate. Additionally, intestinal barrier dysfunction can lead to the translocation of uremic toxins, cytokines, and bacteria into the bloodstream, thereby triggering oxidative stress and chronic inflammation.

Limited clinical evidence exists regarding the relationship between plasma or urine oxalate levels and CKD progression. However, the current findings suggest that hyperoxaluria and hyperoxalemia are associated with CKD progression and adverse outcomes. Therefore, management of the oxalate burden in CKD patients through dietary interventions, modification of risk factors, medication, or DKRT, if indicated, may hold promise in preventing CKD progression and improving cardiovascular outcomes.

Although considerable progress has been made in understanding oxalate’s involvement in CKD, there is still a significant knowledge gap in this area, highlighting the need for further research. Furthermore, there is a lack of awareness and understanding among healthcare providers regarding the importance of managing the oxalate burden in CKD. Enhancing awareness and understanding of the underlying mechanisms of oxalate imbalance in CKD is crucial for identifying at-risk patients, enabling early diagnosis and intervention, and ultimately improving clinical outcomes. To enhance our comprehension of oxalate homeostasis in CKD, a number of crucial research avenues should be pursued (Figure 4).

These research directions will help bridge the knowledge gap, deepen our understanding of oxalate homeostasis in CKD, and pave the way for the development of targeted interventions to manage CKD progression.

## Figures and Tables

**Figure 1 biomedicines-11-01654-f001:**
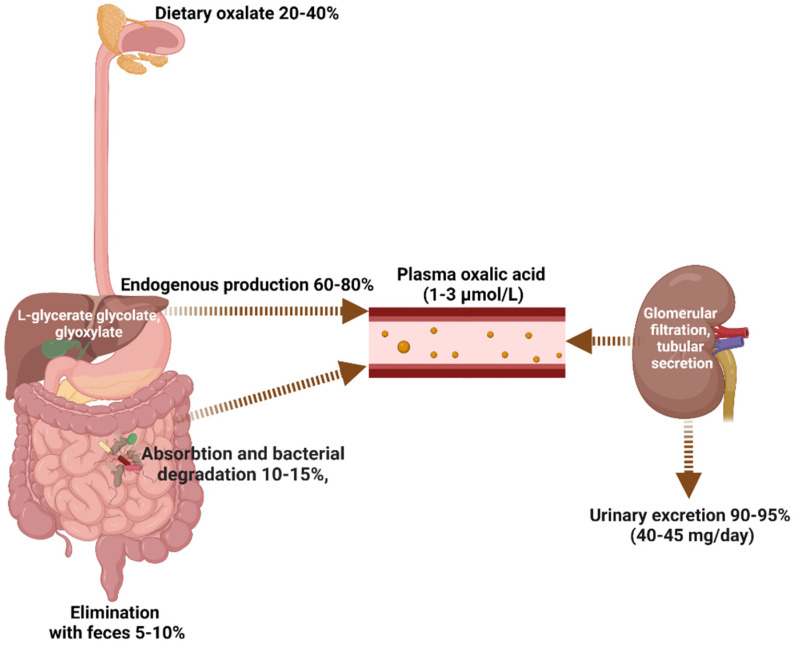
Healthy oxalate homeostasis (created with BioRender.com).

**Figure 2 biomedicines-11-01654-f002:**
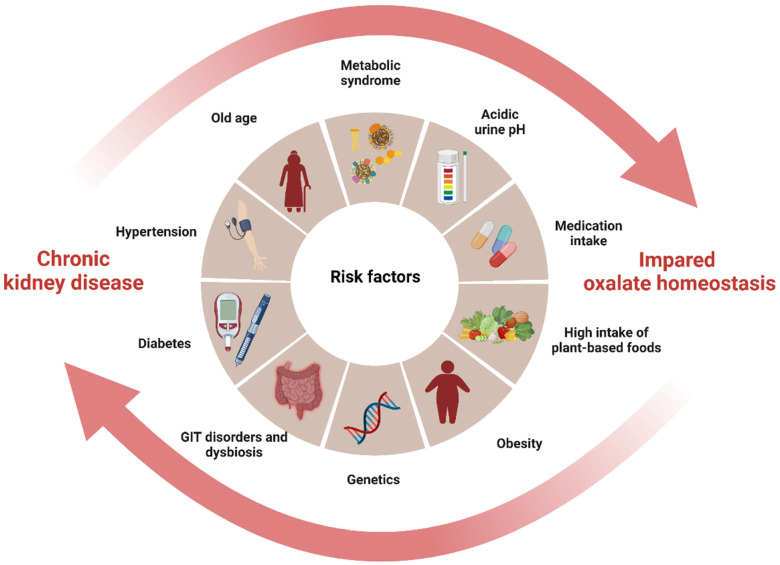
The interplay between impaired oxalate homeostasis and CKD (created with BioRender.com). The diagram illustrates the interplay between impaired oxalate homeostasis and CKD by highlighting the shared risk factors that contribute to both diseases and pointing to their common origin. The cycle formed by these pathological conditions mutually exacerbates each other, leading to an accumulation of oxalate in the body and subsequently increasing the risk of CKD progression.

**Figure 3 biomedicines-11-01654-f003:**
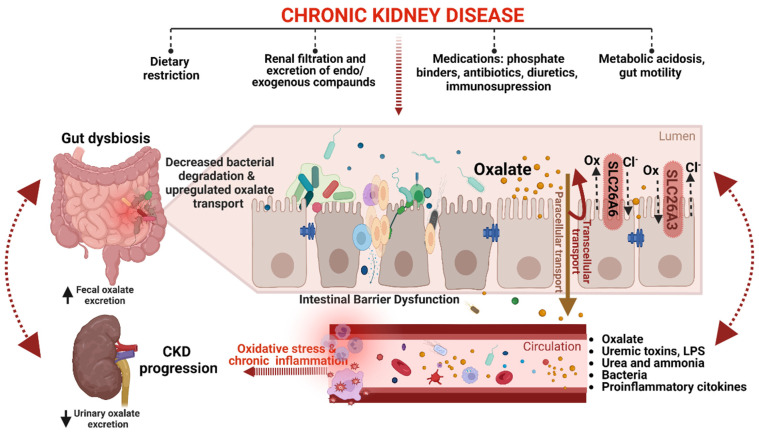
A schematic representation of the gut–kidney axis in CKD oxalate homeostasis (created with BioRender.com). CKD-associated changes in the gut microbiota can modify the abundance of ODB, leading to decreased oxalate degradation in the gut and resulting in hyperoxalemia. Additionally, CKD can affect the expression and activity of transporters, such as SLC26A6, which are involved in oxalate absorption and secretion in the gut, leading to increased fecal oxalate excretion as a compensatory mechanism to maintain oxalate homeostasis. Intestinal barrier dysfunction, on the other hand, can cause the translocation of uremic toxins, proinflammatory cytokines, bacteria, and their byproducts to the circulation, triggering oxidative stress and chronic inflammation. Furthermore, intestinal barrier dysfunction can also affect enteric oxalate handling, thereby exacerbating oxalate-mediated CKD progression. Dietary restriction, medications, gut motility, and metabolic acidosis are some of the factors that can also affect both intestinal barrier dysfunction and oxalate handling in CKD. However, the crosstalk between intestinal barrier dysfunction and enteric oxalate handling in CKD should be an area of future research.

**Figure 4 biomedicines-11-01654-f004:**
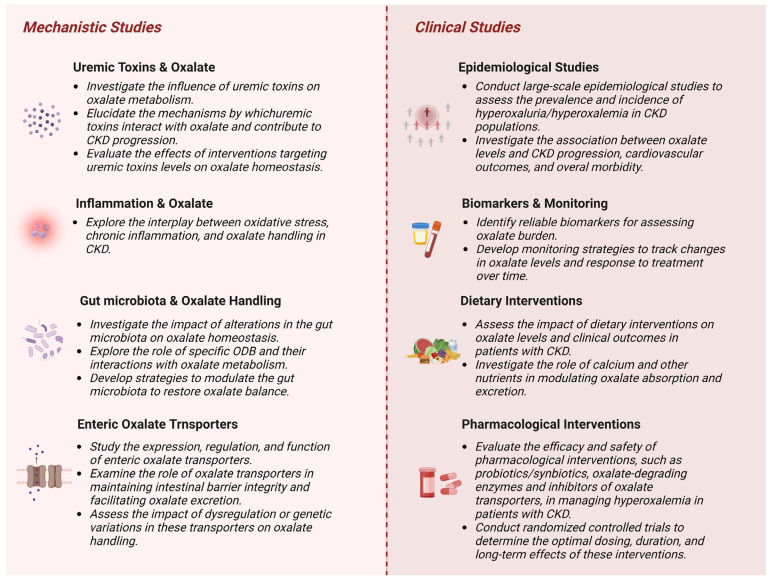
A schematic representation of the proposed research framework for advancing the understanding of oxalate homeostasis in CKD (created with BioRender.com).

## Data Availability

Not applicable.

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
