# Peer review of "Oxalate Homeostasis in Non-Stone-Forming Chronic Kidney Disease: A Review of Key Findings and Perspectives"

_biomedicines, 2023, doi:10.3390/biomedicines11061654_

Round 1
Reviewer 1 Report
Natalia Stepanova has submitted a very comprehensive review of the links between oxalate homeostasis and chronic kidney disease, including a nice description of oxalate formation by the liver and handling in the gut and kidney. The section on medications and their effects on oxalate is unique in highlighting the roles and pitfalls of commonly used medications.
This reviewer has only a few comments
1. Line 121, I believe the author meant SLC26A6, not SLC22A6.
2. Lines 170-173. The author suggests that the high serum oxalate burden in primary hyperoxaluria results in oxalate deposition due to capillary pressure. this reviewer has never seen this mechanism described. The reference does not discuss this mechanism. Why oxalate is deposited iln specific tissues is not entirely clear and it is important to remember that for a long time vascular calcification was thought to be a passive process as well. Is the author aware of any data addressing mechanisms of deposition? Is calcium oxalate deposited on a hydroxyapatite base such as is the case with kidney stones? Does vascular Ca-P deposition in CKD promote oxalate deposition?
3. In some sections, the author describes pathophysiologic phenomena as if they were well-established when that is not entirely the case. For example, lines 206-207, the author states that the imbalanced intake of oxalate and calcium in the diet increases HT risk. Is the author suggesting that this is a cause of HT? or is this just an association? Where abnormal oxalate metabolism and clinical disease occur together should be described as associations unless the author can define a known mechanism. Another example is Lines 289-294 where the author discusses gut disease and abnormal oxalate metabolism as a cause of chronic kidney disease. Again, has this been definitively proven or is this an association?
4. In the section on Oxalate's role in the pathogenesis of CKD, the author discusses multiple mechanisms and observations that have been made. In a couple of instances, the authors specifically state a model where it is studied, such as C57Bl6 mice. It would be important for the author to specify for the other phenomena, what has been seen in pre-clinical model systems and what has been seen in human studies.
5. for the Medication Adjustment section, do the authors have a recommendation on calcium-based binders? For the advanced CKD patient, is the tradeoff of calcium binding to oxalate worth the risk of worsening vascular calcification? Also, the authors might want to state explicitly what the effect of diuretics are on serum oxalate.
Reviewer 2 Report
Review of the article „Oxalate Homeostasis in Non-Stone-Forming Chronic Kidney 2 Disease: A Review of Key Findings and Perspectives“
The abstract section is informative on the subject. The Introduction section is extensive, well-written, and illustrated in Figure 1.
In section 2 regarding less or insolubility of oxalate with Ca, Mg, Zn, Fe. I would like to see a note of solubility for each of them (solubility product Ksp). That is important for the readers of this subject and of clinical importance. (line 144)
I would expand line 219 concerning Uox and acidic pH as it is also clinically significant.
Medication adjustments: It would be necessary for a short comment of Mg citrate as a potential medication as it is given to prevent CaOx stones. Magnesium inhibits calcium oxalate crystallization in human urine and model systems. Magnesium also inhibits the absorption of dietary oxalate from the gut lumen as well as citrate-rich food. Please comment on this therapy in light of MgOx solubility product Ksp. Citrate in urine gives rise to glycosaminoglycans, which serve as calcium binders in urine (doi: 10.1007/s00431-016-2792-9. and for CKD, doi: 10.1016/j.ekir.2016.12.008).
The Figures are very illustrative, and the References are appropriate
Quality of English is satisfactory
Reviewer 3 Report
The paper presents a review of oxalate homeostasis in non-stone-forming chronic kidney disease. The paper is well organized and presents information of high interest. Perhaps the part that should be re-written more clearly, is the one referring to Conclusions. The part of present knowledge should be commented and separated from the studies that should be carried out in the near future, that can be clearly presented schematically. (Minor revision)
